# Near-Optimal Representation Learning for Hierarchical Reinforcement Learning

**Ofir Nachum, Shixiang Gu, Honglak Lee & Sergey Levine**[*]
Google Brain
{ofirnachum,shanegu,honglak,slevine}@google.com

## Abstract

We study the problem of representation learning in goal-conditioned hierarchical reinforcement learning. In such hierarchical structures, a higher-level controller solves tasks by iteratively communicating goals which a lower-level policy is trained to reach. Accordingly, the choice of representation – the mapping of observation space to goal space – is crucial. To study this problem, we develop a notion of sub-optimality of a representation, defined in terms of expected reward of the optimal hierarchical policy using this representation. We derive expressions which bound the sub-optimality and show how these expressions can be translated to representation learning objectives which may be optimized in practice. Results on a number of difficult continuous-control tasks show that our approach to representation learning yields qualitatively better representations as well as quantitatively better hierarchical policies, compared to existing methods.[1][2]

## 1 Introduction

Hierarchical reinforcement learning has long held the promise of extending the successes of existing reinforcement learning (RL) methods (Gu et al., 2017; Schulman et al., 2015; Lillicrap et al., 2015) to more complex, difficult, and temporally extended tasks (Parr & Russell, 1998; Sutton et al., 1999; Barto & Mahadevan, 2003). Recently, *goal-conditioned* hierarchical designs, in which higher-level policies communicate *goals* to lower-levels and lower-level policies are rewarded for reaching states (i.e. observations) which are close to these desired goals, have emerged as an effective paradigm for hierarchical RL (Nachum et al. (2018); Levy et al. (2017); Vezhnevets et al. (2017), inspired by earlier work Dayan & Hinton (1993); Schmidhuber & Wahnsiedler (1993)). In this hierarchical design, *representation learning* is crucial; that is, a *representation function* must be chosen mapping state observations to an abstract space. Goals (desired states) are then specified by the choice of a point in this abstract space.

Previous works have largely studied two ways to choose the representation: learning the representation end-to-end together with the higher- and lower-level policies (Vezhnevets et al., 2017), or using the state space as-is for the goal space (i.e., the goal space is a subspace of the state space) (Nachum et al., 2018; Levy et al., 2017). The former approach is appealing, but in practice often produces poor results (see Nachum et al. (2018) and our own experiments), since the resulting representation is under-defined; i.e., not all possible sub-tasks are expressible as goals in the space. On the other hand, fixing the representation to be the full state means that no information is lost, but this choice is difficult to scale to higher dimensions. For example, if the state observations are entire images, the higher-level must output target images for the lower-level, which can be very difficult.

We instead study how unsupervised objectives can be used to train a representation that is more concise than the full state, but also not as under-determined as in the end-to-end approach. In order to do so in a principled manner, we propose a measure of sub-optimality of a given representation. This measure aims to answer the question: How much does using the learned representation in place of the full representation cause us to lose, in terms of expected reward, against the optimal policy? This question is important, because a useful representation will compress the state, hopefully making

---

[*]Also at UC Berkeley.

[1]See videos at `https://sites.google.com/view/representation-hrl`

[2]Find open-source code at `https://github.com/tensorflow/models/tree/master/research/efficient-hrl`

the learning problem easier. At the same time, the compression might cause the representation to lose information, making the optimal policy impossible to express. It is therefore critical to understand how lossy a learned representation is, not in terms of reconstruction, but in terms of the ability to represent near-optimal policies on top of this representation.

Our main theoretical result shows that, for a particular choice of representation learning objective, we can learn representations for which the return of the hierarchical policy approaches the return of the optimal policy within a bounded error. This suggests that, if the representation is learned with a principled objective, the 'lossy-ness' in the resulting representation should not cause a decrease in overall task performance. We then formulate a representation learning approach that optimizes this bound. We further extend our result to the case of temporal abstraction, where the higher-level controller only chooses new goals at fixed time intervals. To our knowledge, this is the first result showing that hierarchical goal-setting policies with learned representations and temporal abstraction can achieve bounded sub-optimality against the optimal policy. We further observe that the representation learning objective suggested by our theoretical result closely resembles several other recently proposed objectives based on mutual information (van den Oord et al., 2018; Ishmael Belghazi et al., 2018; Hjelm et al., 2018), suggesting an intriguing connection between mutual information and goal representations for hierarchical RL. Results on a number of difficult continuous-control navigation tasks show that our principled representation learning objective yields good qualitative and quantitative performance compared to existing methods.

## 2  FRAMEWORK

Following previous work (Nachum et al., 2018), we consider a two-level hierarchical policy on an MDP $\mathcal{M} = (S, A, R, T)$, in which the higher-level policy modulates the behavior of a lower-level policy by choosing a desired goal state and rewarding the lower-level policy for reaching this state. While prior work has used a sub-space of the state space as goals (Nachum et al., 2018), in more general settings, some type of state representation is necessary. That is, consider a state representation function $f : S \rightarrow \mathbb{R}^d$. A two-level hierarchical policy on $\mathcal{M}$ is composed of a higher-level policy $\pi_{\text{hi}}(g|s)$, where $g \in G = \mathbb{R}^d$ is the goal space, that samples a *high-level action* (or *goal*) $g_t \sim \pi_{\text{hi}}(g|s_t)$ every $c$ steps, for fixed $c$. A non-stationary, goal-conditioned, lower-level policy $\pi_{\text{lo}}(a|s_t, g_t, s_{t+k}, k)$ then translates these high-level actions into low-level actions $a_{t+k} \in A$ for $k \in [0, c-1]$. The process is then repeated, beginning with the higher-level policy selecting another goal according to $s_{t+c}$. The policy $\pi_{\text{lo}}$ is trained using a goal-conditioned reward; e.g. the reward of a transition $g, s, s'$ is $-D(f(s'), g)$, where $D$ is a distance function.

In this work we adopt a slightly different interpretation of the lower-level policy and its relation to $\pi_{\text{hi}}$. Every $c$ steps, the higher-level policy chooses a goal $g_t$ based on a state $s_t$. We interpret this state-goal pair as being mapped to a non-stationary policy $\pi(a|s_{t+k}, k), \pi \in \Pi$, where $\Pi$ denotes the set of all possible $c$-step policies acting on $\mathcal{M}$. We use $\Psi$ to denote this mapping from $S \times G$ to $\Pi$. In other words, on every $c^{\text{th}}$ step, we encounter some state $s_t \in S$. We use the higher-level policy to sample a goal

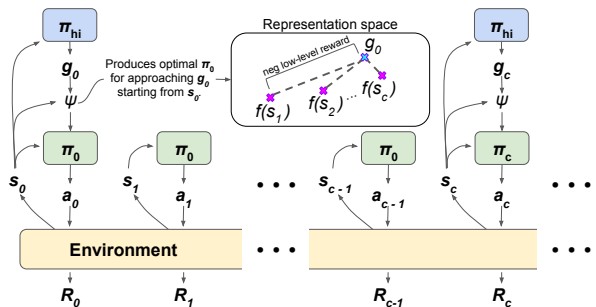

Figure 1: The hierarchical design we consider.

$g_t \sim \pi_{\text{hi}}(g|s_t)$ and translate this to a policy $\pi_t = \Psi(s_t, g_t)$. We then use $\pi_t$ to sample actions $a_{t+k} \sim \pi_t(a|s_{t+k}, k)$ for $k \in [0, c-1]$. The process is then repeated from $s_{t+c}$.

Although the difference in this interpretation is subtle, the introduction of $\Psi$ is crucial for our subsequent analysis. The communication of $g_t$ is no longer as a goal which $\pi_{\text{hi}}$ desires to reach, but rather more precisely, as an identifier to a low-level behavior which $\pi_{\text{hi}}$ desires to induce or activate.

The mapping $\Psi$ is usually expressed as the result of an RL optimization over $\Pi$; e.g.,

$$\Psi(s_t, g) = \arg\max_{\pi \in \Pi} \sum_{k=1}^{c} \gamma^{k-1} \mathbb{E}_{P_\pi(s_{t+k}|s_t)}[-D(f(s_{t+k}), g)], \qquad (1)$$

where we use $P_\pi(s_{t+k}|s_t)$ to denote the probability of being in state $s_{t+k}$ after following $\pi$ for $k$ steps starting from $s_t$. We will consider variations on this low-level objective in later sections. From Equation 1 it is clear how the choice of representation $f$ affects $\Psi$ (albeit indirectly).

We will restrict the environment reward function $R$ to be defined only on states. We use $R_{max}$ to denote the maximal absolute reward: $R_{max} = \sup_S |R(s)|$.

## 3 HIERARCHICAL POLICY SUB-OPTIMALITY

In the previous section, we introduced two-level policies where a higher-level policy $\pi_{hi}$ chooses goals $g$, which are translated to lower-level behaviors via $\Psi$. The introduction of this hierarchy leads to a natural question: How much do we lose by learning $\pi_{hi}$ which is only able to act on $\mathcal{M}$ via $\Psi$? The choice of $\Psi$ restricts the type and number of lower-level behaviors that the higher-level policy can induce. Thus, the optimal policy on $\mathcal{M}$ is potentially not expressible by $\pi_{hi}$. Despite the potential lossy-ness of $\Psi$, can one still learn a hierarchical policy which is near-optimal?

To approach this question, we introduce a notion of sub-optimality with respect to the form of $\Psi$: Let $\pi_{hi}^*(g|s, \Psi)$ be the optimal higher-level policy acting on $G$ and using $\Psi$ as the mapping from $G$ to low-level behaviors. Let $\pi_{hier}^*$ be the corresponding full hierarchical policy on $\mathcal{M}$. We will compare $\pi_{hier}^*$ to an optimal hierarchical policy $\pi^*$ agnostic to $\Psi$. To define $\pi^*$ we begin by introducing an optimal higher-level policy $\pi_{hi}^{**}(\pi|s)$ agnostic to $\Psi$; i.e. every $c$ steps, $\pi_{hi}^{**}$ samples a low-level behavior $\pi \in \Pi$ which is applied to $\mathcal{M}$ for the following $c$ steps. In this way, $\pi_{hi}^{**}$ may express all possible low-level behaviors. We then denote $\pi^*$ as the full hierarchical policy resulting from $\pi_{hi}^{**}$.

We would like to compare $\pi_{hier}^*$ to $\pi^*$. A natural and common way to do so is in terms of state values. Let $V^\pi(s)$ be the future value achieved by a policy $\pi$ starting at state $s$. We define the *sub-optimality* of $\Psi$ as

$$\text{SubOpt}(\Psi) = \sup_{s \in S} V^{\pi^*}(s) - V^{\pi_{hier}^*}(s). \tag{2}$$

The state values $V^{\pi_{hier}^*}(s)$ are determined by the form of $\Psi$, which is in turn determined by the choice of representation $f$. However, none of these relationships are direct. It is unclear how a change in $f$ will result in a change to the sub-optimality. In the following section, we derive a series of bounds which establish a more direct relationship between $\text{SubOpt}(\Psi)$ and $f$. Our main result will show that if one defines $\Psi$ as a slight modification of the traditional objective given in Equation 1, then one may translate sub-optimality of $\Psi$ to a practical representation learning objective for $f$.

## 4 GOOD REPRESENTATIONS LEAD TO BOUNDED SUB-OPTIMALITY

In this section, we provide proxy expressions that bound the sub-optimality induced by a specific choice of $\Psi$. Our main result is Claim 4, which connects the sub-optimality of $\Psi$ to both goal-conditioned policy objectives (i.e., the objective in 1) and representation learning (i.e., an objective for the function $f$).

### 4.1 SINGLE-STEPS ($c = 1$) AND DETERMINISTIC POLICIES

For ease of presentation, we begin by presenting our results in the restricted case of $c = 1$ and deterministic lower-level policies. In this setting, the class of low-level policies $\Pi$ may be taken to be simply $A$, where $a \in \Pi$ corresponds to a policy which always chooses action $a$. There is no temporal abstraction: The higher-level policy chooses a high-level action $g \in G$ at every step, which is translated via $\Psi$ to a low-level action $a \in A$. Our claims are based on quantifying how many of the possible low-level behaviors (i.e., all possible state to state transitions) can be produced by $\Psi$ for different choices of $g$. To quantify this, we make use of an auxiliary *inverse goal model* $\varphi(s, a)$, which aims to predict which goal $g$ will cause $\Psi$ to yield an action $\tilde{a} = \Psi(s, g)$ that induces a next state distribution $P(s'|s, \tilde{a})$ similar to $P(s'|s, a)$.[3] We have the following theorem, which bounds the sub-optimality in terms of total variation divergences between $P(s'|s, a)$ and $P(s'|s, \tilde{a})$:

**Theorem 1.** *If there exists $\varphi : S \times A \to G$ such that,*

$$\sup_{s \in S, a \in A} D_{\text{TV}}(P(s'|s, a) || P(s'|s, \Psi(s, \varphi(s, a)))) \leq \epsilon, \tag{3}$$

*then* $\text{SubOpt}(\Psi) \leq C\epsilon$*, where* $C = \frac{2\gamma}{(1-\gamma)^2} R_{max}$*.*

---

[3]In a deterministic, $c = 1$ setting, $\varphi$ may be seen as a state-conditioned action abstraction mapping $A \to G$.

*Proof.* See Appendices A and B for all proofs.

Theorem 1 allows us to bound the sub-optimality of $\Psi$ in terms of how recoverable the effect of any action in $A$ is, in terms of transition to the next state. One way to ensure that effects of actions in $A$ are recoverable is to have an invertible $\Psi$. That is, if there exists $\varphi : S \times A \to G$ such that $\Psi(s, \varphi(s, a)) = a$ for all $s, a$, then the sub-optimality of $\Psi$ is 0.

However, in many cases it may not be desirable or feasible to have an invertible $\Psi$. Looking back at Theorem 1, we emphasize that its statement requires only the *effect* of any action to be recoverable. That is, for any $s, \in S, a \in A$, we require only that there exist some $g \in G$ (given by $\varphi(s, a)$) which yields a similar next-state distribution. To this end, we have the following claim, which connects the sub-optimality of $\Psi$ to both representation learning and the form of the low-level objective.

**Claim 2.** *Let $\rho(s)$ be a prior and $f, \varphi$ be so that, for $K(s'|s, a) \propto \rho(s') \exp(-D(f(s'), \varphi(s, a)))$,[4]*

$$\sup_{s \in S, a \in A} D_{\mathrm{KL}}(P(s'|s, a) || K(s'|s, a)) \le \epsilon^2/8. \tag{4}$$

*If the low-level objective is defined as*

$$\Psi(s, g) = \arg\max_{a \in A} \mathbb{E}_{P(s'|s,a)}[-D(f(s'), g) + \log \rho(s') - \log P(s'|s, a)], \tag{5}$$

*then the sub-optimality of $\Psi$ is bounded by $C\epsilon$.*

We provide an intuitive explanation of the statement of Claim 2. First, consider that the distribution $K(s'|s, a)$ appearing in Equation 4 may be interpreted as a dynamics model determined by $f$ and $\varphi$. By bounding the difference between the true dynamics $P(s'|s, a)$ and the dynamics $K(s'|s, a)$ implied by $f$ and $\varphi$, Equation 4 states that the representation $f$ should be chosen in such a way that dynamics in representation space are roughly given by $\varphi(s, a)$. This is essentially a representation learning objective for choosing $f$, and in Section 5 we describe how to optimize it in practice.

Moving on to Equation 5, we note that the form of $\Psi$ here is *only slightly* different than the one-step form of the standard goal-conditioned objective in Equation 1.[5] Therefore, all together Claim 2 establishes a deep connection between representation learning (Equation 4), goal-conditioned policy learning (Equation 5), and sub-optimality.

## 4.2 TEMPORAL ABSTRACTION ($c \ge 1$) AND GENERAL POLICIES

We now move on to presenting the same results in the fully general, temporally abstracted setting, in which the higher-level policy chooses a high-level action $g \in G$ every $c$ steps, which is transformed via $\Psi$ to a $c$-step lower-level behavior policy $\pi \in \Pi$. In this setting, the auxiliary inverse goal model $\varphi(s, \pi)$ is a mapping from $S \times \Pi$ to $G$ and aims to predict which goal $g$ will cause $\Psi$ to yield a policy $\tilde{\pi} = \Psi(s, g)$ that induces future state distributions $P_{\tilde{\pi}}(s_{t+k}|s_t)$ similar to $P_{\pi}(s_{t+k}|s_t)$, for $k \in [1, c]$. We weight the divergences between the distributions by weights $w_k = 1$ for $k < c$ and $w_k = (1 - \gamma)^{-1}$ for $k = c$. We denote $\overline{w} = \sum_{k=1}^{c} \gamma^{k-1} w_k$. The analogue to Theorem 1 is as follows:

**Theorem 3.** *Consider a mapping $\varphi : S \times \Pi \to G$ and define $\epsilon_k : S \times \Pi \to \mathbb{R}$ for $k \in [1, c]$ as,*

$$\epsilon_k(s_t, \pi) = D_{\mathrm{TV}}(P_{\pi}(s_{t+k}|s_t) || P_{\Psi(s_t, \varphi(s_t, \pi))}(s_{t+k}|s_t)). \tag{6}$$

*If*

$$\sup_{s_t \in S, \pi \in \Pi} \frac{1}{\overline{w}} \sum_{k=1}^{c} \gamma^{k-1} w_k \epsilon_k(s_t, \pi) \le \epsilon, \tag{7}$$

*then $\mathrm{SubOpt}(\Psi) \le C\epsilon$, where $C = \frac{2\gamma}{1-\gamma^c} R_{max}\overline{w}$.*

For the analogue to Claim 2, we simply replace the single-step KL divergences and low-level rewards with a discounted weighted sum thereof:

---

[4] $K$ may be interpreted as the conditional $P(\text{state} = s'|\text{repr} = \varphi(s, a))$ of the joint distribution $P(\text{state} = s')P(\text{repr} = z|\text{state} = s') = \rho(s') \exp(-D(f(s'), z))/Z$ for normalization constant $Z$.

[5] As evident in the proof of this Claim (see Appendix), the form of Equation 5 is specifically designed so that it corresponds to a KL between $P(s'|s, a)$ and a distribution $U(s'|s, g) \propto \rho(s') \exp(-D(f(s'), g))$.

**Claim 4.** *Let $\rho(s)$ be a prior over $S$. Let $f, \varphi$ be such that,*

$$\sup_{s_t \in S, \pi \in \Pi} \frac{1}{\overline{w}} \sum_{k=1}^{c} \gamma^{k-1} w_k D_{\mathrm{KL}}(P_\pi(s_{t+k}|s_t) || K(s_{t+k}|s_t, \pi)) \leq \epsilon^2/8, \tag{8}$$

*where $K(s_{t+k}|s_t, \pi) \propto \rho(s_{t+k}) \exp(-D(f(s_{t+k}), \varphi(s_t, \pi)))$.*

*If the low-level objective is defined as*

$$\Psi(s_t, g) = \arg\max_{\pi \in \Pi} \sum_{k=1}^{c} \gamma^{k-1} w_k \mathbb{E}_{P_\pi(s_{t+k}|s_t)}[-D(f(s_{t+k}), g) + \log \rho(s_{t+k}) - \log P_\pi(s_{t+k}|s_t)], \tag{9}$$

*then the sub-optimality of $\Psi$ is bounded by $C\epsilon$.*

Claim 4 is the main theoretical contribution of our work. As in the previous claim, we have a strong statement, saying that if the low-level objective is defined as in Equation 9, then minimizing the sub-optimality may be done by optimizing a representation learning objective based on Equation 8. We emphasize that Claim 4 applies to any class of low-level policies $\Pi$, including either closed-loop or open-loop policies.

## 5 LEARNING

We now have the mathematical foundations necessary to learn representations that are provably good for use in hierarchical RL. We begin by elaborating on how we translate Equation 8 into a practical training objective for $f$ and auxiliary $\varphi$ (as well as a practical parameterization of policies $\pi$ as input to $\varphi$). We then continue to describe how one may train a lower-level policy to match the objective presented in Equation 9. In this way, we may learn $f$ and lower-level policy to directly optimize a bound on the sub-optimality of $\Psi$. A pseudocode of the full algorithm is presented in the Appendix (see Algorithm 1).

### 5.1 LEARNING GOOD REPRESENTATIONS

Consider a representation function $f_\theta : S \to \mathbb{R}^d$ and an auxiliary function $\varphi_\theta : S \times \Pi \to \mathbb{R}^d$, parameterized by vector $\theta$. In practice, these are separate neural networks: $f_{\theta_1}, \varphi_{\theta_2}, \theta = [\theta_1, \theta_2]$.

While the form of Equation 8 suggests to optimize a supremum over all $s_t$ and $\pi$, in practice we only have access to a replay buffer which stores experience $s_0, a_0, s_1, a_1, \ldots$ sampled from our hierarchical behavior policy. Therefore, we propose to choose $s_t$ sampled uniformly from the replay buffer and use the subsequent $c$ actions $a_{t:t+c-1}$ as a representation of the policy $\pi$, where we use $a_{t:t+c-1}$ to denote the sequence $a_t, \ldots, a_{t+c-1}$. Note that this is equivalent to setting the set of candidate policies $\Pi$ to $A^c$ (i.e., $\Pi$ is the set of $c$-step, deterministic, open-loop policies). This choice additionally simplifies the possible structure of the function approximator used for $\varphi_\theta$ (a standard neural net which takes in $s_t$ and $a_{t:t+c-1}$). Our proposed representation learning objective is thus,

$$J(\theta) = \mathbb{E}_{s_t, a_{t:t+c-1} \sim \text{replay}}[J(\theta, s_t, a_{t:t+c-1})], \tag{10}$$

where $J(\theta, s_t, a_{t:t+c-1})$ will correspond to the inner part of the supremum in Equation 8.

We now define the inner objective $J(\theta, s_t, a_{t:t+c-1})$. To simplify notation, we use $E_\theta(s', s, \pi) = \exp(-D(f_\theta(s'), \varphi_\theta(s, \pi)))$ and use $K_\theta(s'|s, \pi)$ as the distribution over $S$ such that $K_\theta(s'|s, \pi) \propto \rho(s')E_\theta(s', s, \pi)$. Equation 8 suggests the following learning objective on each $s_t, \pi \equiv a_{t:t+c-1}$:

$$J(\theta, s_t, \pi) = \sum_{k=1}^{c} \gamma^{k-1} w_k D_{\mathrm{KL}}(P_\pi(s_{t+k}|s_t) || K_\theta(s_{t+k}|s_t, \pi)) \tag{11}$$

$$= B + \sum_{k=1}^{c} -\gamma^{k-1} w_k \mathbb{E}_{P_\pi(s_{t+k}|s_t)} \left[ \log K_\theta(s_{t+k}|s_t, \pi) \right] \tag{12}$$

$$= B + \sum_{k=1}^{c} -\gamma^{k-1} w_k \mathbb{E}_{P_\pi(s_{t+k}|s_t)} \left[ \log E_\theta(s_{t+k}, s_t, \pi) \right] + \gamma^{k-1} w_k \log \mathbb{E}_{\tilde{s} \sim \rho} \left[ E_\theta(\tilde{s}, s_t, \pi) \right], \tag{13}$$

where $B$ is a constant. The gradient with respect to $\theta$ is then,

$$\sum_{k=1}^{c} -\gamma^{k-1} w_k \mathbb{E}_{P_\pi(s_{t+k}|s_t)} \left[ \nabla_\theta \log E_\theta(s_{t+k}, s_t, \pi) \right] + \gamma^{k-1} w_k \frac{\mathbb{E}_{\tilde{s} \sim \rho} \left[ \nabla_\theta E_\theta(\tilde{s}, s_t, \pi) \right]}{\mathbb{E}_{\tilde{s} \sim \rho} \left[ E_\theta(\tilde{s}, s_t, \pi) \right]} \tag{14}$$

The first term of Equation 14 is straightforward to estimate using experienced $s_{t+1:t+k}$. We set $\rho$ to be the replay buffer distribution, so that the numerator of the second term is also straightforward. We approximate the denominator of the second term using a mini-batch $\widetilde{S}$ of states independently sampled from the replay buffer:

$$\mathbb{E}_{\tilde{s} \sim \rho} \left[ E_\theta(\tilde{s}, s_t, \pi) \right] \approx |\widetilde{S}|^{-1} \sum_{\tilde{s} \in \widetilde{S}} E_\theta(\tilde{s}, s_t, \pi). \tag{15}$$

This completes the description of our representation learning algorithm.

**Connection to Mutual Information Estimators.** The form of the objective we optimize (i.e. Equation 13) is very similar to mutual information estimators, mostly CPC (van den Oord et al., 2018). Indeed, one may interpret our objective as maximizing a mutual information $MI(s_{t+k}; s_t, \pi)$ via an energy function given by $E_\theta(s_{t+k}, s_t, \pi)$. The main differences between our approach and these previous proposals are as follows: (1) Previous approaches maximize a mutual information $MI(s_{t+k}; s_t)$ agnostic to actions or policy. (2) Previous approaches suggest to define the energy function as $\exp(f(s_{t+k})^T M_k f(s_t))$ for some matrix $M_k$, whereas our energy function is based on the distance $D$ used for low-level reward. (3) Our approach is provably good for use in hierarchical RL, and hence our theoretical results may justify some of the good performance observed by others using mutual information estimators for representation learning. Different approaches to translating our theoretical findings to practical implementations may yield objectives more or less similar to CPC, some of which perform better than others (see Appendix D).

### 5.2 LEARNING A LOWER-LEVEL POLICY

Equation 9 suggests to optimize a policy $\pi_{s_t,g}(a|s_{t+k}, k)$ for every $s_t, g$. This is equivalent to the parameterization $\pi_{\text{lo}}(a|s_t, g, s_{t+k}, k)$, which is standard in goal-conditioned hierarchical designs. Standard RL algorithms may be employed to maximize the low-level reward implied by Equation 9:

$$- D(f(s_{t+k}), g) + \log \rho(s_{t+k}) - \log P_\pi(s_{t+k}|s_t), \tag{16}$$

weighted by $w_k$ and where $\pi$ corresponds to $\pi_{\text{lo}}$ when the state $s_t$ and goal $g$ are fixed. While the first term of Equation 16 is straightforward to compute, the log probabilities $\log \rho(s_{t+k}), \log P_\pi(s_{t+k}|s_t)$ are in general unknown. To approach this issue, we take advantage of the representation learning objective for $f, \varphi$. When $f, \varphi$ are optimized as dictated by Equation 8, we have

$$\log P_\pi(s_{t+k}|s_t) \approx \log \rho(s_{t+k}) - D(f(s_{t+k}), \varphi(s_t, \pi)) - \log \mathbb{E}_{\tilde{s} \sim \rho}[E(\tilde{s}, s_t, \pi)]. \tag{17}$$

We may therefore approximate the low-level reward as

$$- D(f(s_{t+k}), g) + D(f(s_{t+k}), \varphi(s_t, \pi)) + \log \mathbb{E}_{\tilde{s} \sim \rho}[E(\tilde{s}, s_t, \pi)]. \tag{18}$$

As in Section 5.1, we use the sampled actions $a_{t:t+c-1}$ to represent $\pi$ as input to $\varphi$. We approximate the third term of Equation 18 analogously to Equation 15. Note that this is a slight difference from standard low-level rewards, which use only the first term of Equation 18 and are unweighted.

## 6 RELATED WORK

Representation learning for RL has a rich and diverse existing literature, often interpreted as an *abstraction* of the original MDP. Previous works have interpreted the hierarchy introduced in hierarchical RL as an MDP abstraction of state, action, and temporal spaces (Sutton et al., 1999; Dietterich, 2000; Thomas & Barto, 2012; Bacon et al., 2017). In goal-conditioned hierarchical designs, although the representation is learned on states, it is in fact a form of action abstraction (since goals $g$ are high-level actions). While previous successful applications of goal-conditioned hierarchical designs have either learned representations naively end-to-end (Vezhnevets et al., 2017), or not learned them at all (Levy et al., 2017; Nachum et al., 2018), we take a principled approach to representation learning in hierarchical RL, translating a bound on sub-optimality to a practical learning objective.

Bounding sub-optimality in abstracted MDPs has a long history, from early work in theoretical analysis on approximations to dynamic programming models (Whitt, 1978; Bertsekas & Castanon, 1989). Extensive theoretical work on state abstraction, also known as state aggregation or model minimization, has been done in both operational research (Rogers et al., 1991; Van Roy, 2006) and RL (Dean & Givan, 1997; Ravindran & Barto, 2002; Abel et al., 2017). Notably, Li et al. (2006)

introduce a formalism for categorizing classic work on state abstractions such as bisimulation (Dean & Givan, 1997) and homomorphism (Ravindran & Barto, 2002) based on what information is preserved, which is similar in spirit to our approach. Exact state abstractions (Li et al., 2006) incur no performance loss (Dean & Givan, 1997; Ravindran & Barto, 2002), while their approximate variants generally have bounded sub-optimality (Bertsekas & Castanon, 1989; Dean & Givan, 1997; Sorg & Singh, 2009; Abel et al., 2017). While some of the prior work also focuses on learning state abstractions (Li et al., 2006; Sorg & Singh, 2009; Abel et al., 2017), they often exclusively apply to simple MDP domains as they rely on techniques such as state partitioning or Q-value based aggregation, which are difficult to scale to our experimented domains. Thus, the key differentiation of our work from these prior works is that we derive bounds which may be translated to practical representation learning objectives. Our impressive results on difficult continuous-control, high-dimensional domains is a testament to the potential impact of our theoretical findings.

Lastly, we note the similarity of our representation learning algorithm to recently introduced scalable mutual information maximization objectives such as CPC (van den Oord et al., 2018) and MINE (Ishmael Belghazi et al., 2018). This is not a surprise, since maximizing mutual information relates closely with maximum likelihood learning of energy-based models, and our bounds effectively correspond to bounds based on model-based predictive errors, a basic family of bounds in representation learning in MDPs (Sorg & Singh, 2009; Brunskill & Li, 2014; Abel et al., 2017). Although similar information theoretic measures have been used previously for exploration in RL (Still & Precup, 2012), to our knowledge, no prior work has connected these mutual information estimators to representation learning in hierarchical RL, and ours is the first to formulate theoretical guarantees on sub-optimality of the resulting representations in such a framework.

## 7 EXPERIMENTS

We evaluate our proposed representation learning objective compared to a number of baselines:

- XY: The oracle baseline which uses the $x, y$ position of the agent as the representation.

- VAE: A variational autoencoder (Kingma & Welling, 2013) on raw observations.

- E2C: Embed to control (Watter et al., 2015). A method which uses variational objectives to train a representation of states and actions which have locally linear dynamics.

- E2E: End-to-end learning of the representation. The representation is fed as input to the higher-level policy and learned using gradients from the RL objective.

- Whole obs: The raw observation is used as the representation. No representation learning. This is distinct from Nachum et al. (2018), in which a subset of the observation space was pre-determined for use as the goal space.

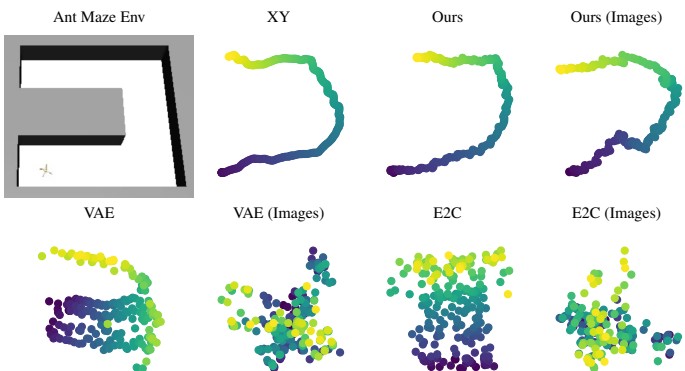

Figure 2: Learned representations (2D embeddings) of our method and a number of variants on a MuJoCo Ant Maze environment, with color gradient based on episode time-step (black for beginning of episode, yellow for end). The ant travels from beginning to end of a ⊃-shaped corridor along an $x, y$ trajectory shown under XY. Without any supervision, our method is able to deduce this near-ideal representation, even when the raw observation is given as a top-down image. Other approaches are unable to properly recover a good representation.

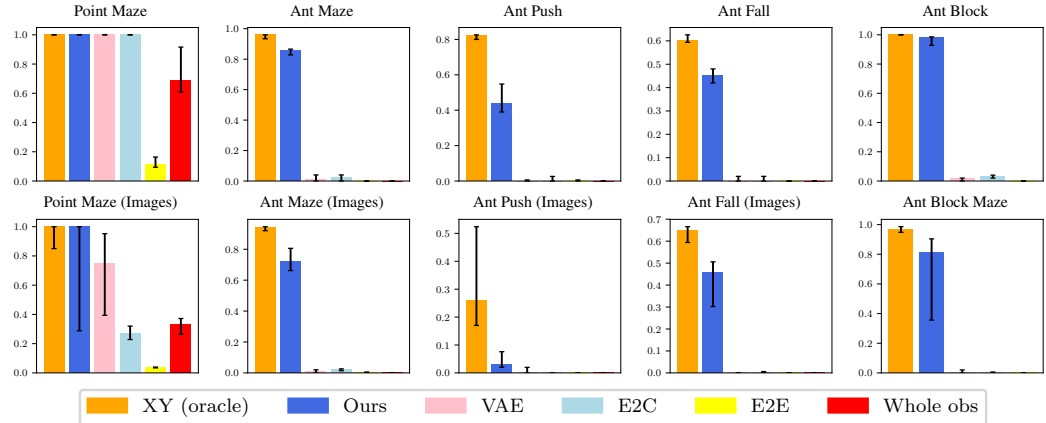

Figure 3: Results of our method and a number of variants on a suite of tasks in 10M steps of training, plotted according to median over 10 trials with $30^{\text{th}}$ and $70^{\text{th}}$ percentiles. We find that outside of simple point environments, our method is the only one which can approach the performance of oracle $x, y$ representations. These results show that our method can be successful, even when the representation is learned online concurrently while learning a hierarchical policy.

We evaluate on the following continuous-control MuJoCo (Todorov et al., 2012) tasks (see Appendix C for details):

- Ant (or Point) Maze: An ant (or point mass) must navigate a ⊃-shaped corridor.

- Ant Push: An ant must push a large block to the side to reach a point behind it.

- Ant Fall: An ant must push a large block into a chasm so that it may walk over it to the other side without falling.

- Ant Block: An ant must push a small block to various locations in a square room.

- Ant Block Maze: An ant must push a small block through a ⊃-shaped corridor.

In these tasks, the raw observation is the agent's $x, y$ coordinates and orientation as well as local coordinates and orientations of its limbs. In the Ant Block and Ant Block Maze environments we also include the $x, y$ coordinates and orientation of the block. We also experiment with more difficult raw representations by replacing the $x, y$ coordinates of the agent with a low-resolution $5 \times 5 \times 3$ top-down image of the agent and its surroundings. These experiments are labeled 'Images'.

For the baseline representation learning methods which are agnostic to the RL training (VAE and E2C), we provide comparative qualitative results in Figure 2. These representations are the result

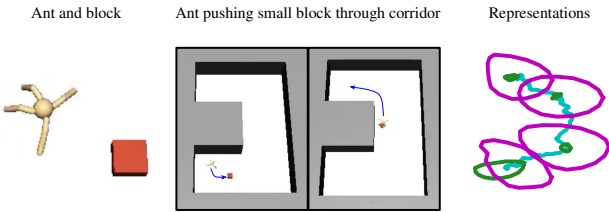

Figure 4: We investigate importance of various observation coordinates in learned representations on a difficult block-moving task. In this task, a simulated robotic ant must move a small red block from beginning to end of a ⊃-shaped corridor. Observations include both ant and block $x, y$ coordinates. We show the trajectory of the learned representations on the right (cyan). At four time steps, we also plot the resulting representations after perturbing the observation's ant coordinates (green) or the observation's block coordinates (magenta). The learned representations put a greater emphasis (i.e., higher sensitivity) on the block coordinates, which makes sense for this task as the external reward is primarily determined by the position of the block.

of taking a trained policy, fixing it, and using its sampled experience to learn 2D representations of the raw observations. We find that our method can successfully deduce the underlying near-optimal $x, y$ representation, even when the raw observation is given as an image.

We provide quantitative results in Figure 3. In these experiments, the representation is learned concurrently while learning a full hierarchical policy (according to the procedure in Nachum et al. (2018)). Therefore, this setting is especially difficult since the representation learning must learn good representations even when the behavior policy is very far from optimal. Accordingly, we find that most baseline methods completely fail to make any progress. Only our proposed method is able to approach the performance of the XY oracle.

For the 'Block' environments, we were curious what our representation learning objective would learn, since the $x, y$ coordinate of the agent is not the only near-optimal representation. For example, another suitable representation is the $x, y$ coordinates of the small block. To investigate this, we plotted (Figure 4) the trajectory of the learned representations of a successful policy (cyan), along with the representations of the same observations with agent $x, y$ perturbed (green) or with block $x, y$ perturbed (magenta). We find that the learned representations greatly emphasize the block $x, y$ coordinates over the agent $x, y$ coordinates, although in the beginning of the episode, there is a healthy mix of the two.

## 8   CONCLUSION

We have presented a principled approach to representation learning in hierarchical RL. Our approach is motivated by the desire to achieve maximum possible return, hence our notion of sub-optimality is in terms of optimal state values. Although this notion of sub-optimality is intractable to optimize directly, we are able to derive a mathematical relationship between it and a specific form of representation learning. Our resulting representation learning objective is practical and achieves impressive results on a suite of high-dimensional, continuous-control tasks.

### ACKNOWLEDGMENTS

We thank Bo Dai, Luke Metz, and others on the Google Brain team for insightful comments and discussions.

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

## A   PROOF OF THEOREM 3 (GENERALIZATION OF THEOREM 1)

Consider the sub-optimality with respect to a specific state $s_0$, $V^{\pi^*}(s_0) - V^{\pi^*_{\text{hier}}}(s_0)$. Recall that $\pi^*$ is the hierarchical result of a policy $\pi^{**}_{\text{hi}} : S \to \Delta(\Pi)$, and note that $\pi^{**}_{\text{hi}}$ may be assumed to be deterministic due to the Markovian nature of $\mathcal{M}$. We may use the mapping $\varphi$ to transform $\pi^{**}_{\text{hi}}$ to a high-level policy $\pi_{\text{hi}}$ on $G$ and using the mapping $\Psi$:

$$g \sim \pi_{\text{hi}}(-|s) \equiv \pi \sim \pi^{**}_{\text{hi}}(-|s), g := \varphi(s, \pi). \tag{19}$$

Let $\pi_{\text{hier}}$ be the corresponding hierarchical policy. We will bound the quantity $V^{\pi^*}(s_0) - V^{\pi_{\text{hier}}}(s_0)$, which will bound $V^{\pi^*}(s_0) - V^{\pi^*_{\text{hier}}}(s_0)$. We follow logic similar to Achiam et al. (2017) and begin by bounding the total variation divergence between the $\gamma$-discounted state visitation frequencies of the two policies.

Denote the $k$-step state transition distributions using either $\pi^*$ or $\pi_{\text{hier}}$ as,

$$P^k_*(s'|s) = P_{\pi^{**}_{\text{hi}}(s)}(s_{t+k} = s'|s_t = s), \tag{20}$$

$$P^k_{\text{hier}}(s'|s) = P_{\Psi(s, \pi_{\text{hi}}(s))}(s_{t+k} = s'|s_t = s), \tag{21}$$

for $k \in [1, c]$. Considering $P_*, P_{\text{hier}}$ as linear operators, we may express the state visitation frequencies $d_*, d_{\text{hier}}$ of $\pi^*, \pi_{\text{hier}}$, respectively, as

$$d_* = (1 - \gamma)A_*(I - \gamma^c P^c_*)^{-1}\mu, \tag{22}$$

$$d_{\text{hier}} = (1 - \gamma)A_{\text{hier}}(I - \gamma^c P^c_{\text{hier}})^{-1}\mu, \tag{23}$$

where $\mu$ is a Dirac $\delta$ distribution centered at $s_0$ and

$$A_* = I + \sum_{k=1}^{c-1} \gamma^k P^k_*, \tag{24}$$

$$A_{\text{hier}} = I + \sum_{k=1}^{c-1} \gamma^k P^k_{\text{hier}}. \tag{25}$$

We will use $d^c_*, d^c_{\text{hier}}$ to denote the every-$c$-steps $\gamma$-discounted state frequencies of $\pi^*, \pi_{\text{hier}}$; i.e.,

$$d^c_* = (1 - \gamma^c)(I - \gamma^c P^c_*)^{-1}\mu, \tag{26}$$

$$d^c_{\text{hier}} = (1 - \gamma^c)(I - \gamma^c P^c_{\text{hier}})^{-1}\mu. \tag{27}$$

By the triangle inequality, we have the following bound on the total variation divergence $|d_{\text{hier}} - d_*|$:

$$|d_{\text{hier}} - d_*| \leq (1 - \gamma)|A_{\text{hier}}(I - \gamma^c P^c_{\text{hier}})^{-1}\mu - A_{\text{hier}}(I - \gamma^c P^c_*)^{-1}\mu|$$
$$+ (1 - \gamma)|A_{\text{hier}}(I - \gamma^c P^c_*)^{-1}\mu - A_*(I - \gamma^c P^c_*)^{-1}\mu|. \tag{28}$$

We begin by attacking the first term of Equation 28. We note that

$$|A_{\text{hier}}| \leq |I| + \sum_{k=1}^{c-1} \gamma^k |P^k_{\text{hier}}| = \frac{1 - \gamma^c}{1 - \gamma}. \tag{29}$$

Thus the first term of Equation 28 is bounded by

$$(1 - \gamma^c)|(I - \gamma^c P^c_{\text{hier}})^{-1}\mu - (I - \gamma^c P^c_*)^{-1}\mu|$$
$$= (1 - \gamma^c)|(I - \gamma^c P^c_{\text{hier}})^{-1}((I - \gamma^c P^c_*) - (I - \gamma^c P^c_{\text{hier}}))(I - \gamma^c P^c_*)^{-1}\mu|$$
$$= \gamma^c|(I - \gamma^c P^c_{\text{hier}})^{-1}(P^c_{\text{hier}} - P^c_*)d^c_*|. \tag{30}$$

By expressing $(I - \gamma^c P^c_{\text{hier}})^{-1}$ as a geometric series and employing the triangle inequality, we have $|(I - \gamma^c P^c_{\text{hier}})^{-1}| \leq (1 - \gamma^c)^{-1}$, and we thus bound the whole quantity (30) by

$$\gamma^c(1 - \gamma^c)^{-1}|(P^c_{\text{hier}} - P^c_*)d^c_*|. \tag{31}$$

We now move to attack the second term of Equation 28. We may express this term as

$$(1 - \gamma)(1 - \gamma^c)^{-1}|(A_{\text{hier}} - A_*)d^c_*|. \tag{32}$$

Furthermore, by the triangle inequality we have

$$|(A_{\text{hier}} - A_*)d_*^c| \leq \sum_{k=1}^{c-1} \gamma^k |(P_{\text{hier}}^k - P_*^k)d_*^c|. \tag{33}$$

Therefore, recalling $w_k = 1$ for $k < c$ and $w_k = (1-\gamma)^{-1}$ for $k = c$, we may bound the total variation of the state visitation frequencies as

$$|d_{\text{hier}} - d_*| \leq \gamma(1-\gamma)(1-\gamma^c)^{-1} \sum_{k=1}^{c} \gamma^{k-1} w_k |(P_{\text{hier}}^k - P_*^k)d_*^c| \tag{34}$$

$$= 2\gamma(1-\gamma)(1-\gamma^c)^{-1} \sum_{k=1}^{c} \gamma^{k-1} w_k \mathbb{E}_{s \sim d_*^c}[D_{\text{TV}}(P_*^k(s'|s)||P_{\text{hier}}^k(s'|s))] \tag{35}$$

$$= 2\gamma(1-\gamma)(1-\gamma^c)^{-1} \mathbb{E}_{s \sim d_*^c}\left[\sum_{k=1}^{c} \gamma^{k-1} w_k D_{\text{TV}}(P_*^k(s'|s)||P_{\text{hier}}^k(s'|s))\right]. \tag{36}$$

By condition 7 of Theorem 3 we have,

$$|d_{\text{hier}} - d_*| \leq 2\gamma(1-\gamma)(1-\gamma^c)^{-1} \overline{w}\epsilon \tag{37}$$

We now move to considering the difference in values. We have

$$V^{\pi^*}(s_0) = (1-\gamma)^{-1} \int_S d_*(s) R(s) \, ds, \tag{38}$$

$$V^{\pi_{\text{hier}}}(s_0) = (1-\gamma)^{-1} \int_S d_{\text{hier}}(s) R(s) \, ds. \tag{39}$$

Therefore, we have

$$|V^{\pi^*}(s_0) - V^{\pi_{\text{hier}}}(s_0)| \leq (1-\gamma)^{-1} R_{max} |d_{\text{hier}} - d_*| \tag{40}$$

$$\leq \frac{2\gamma}{1-\gamma^c} R_{max} \overline{w}\epsilon, \tag{41}$$

as desired.

## B    PROOF OF CLAIM 4 (GENERALIZATION OF CLAIM 2)

Consider a specific $s_t, \pi$. Let $K(s'|s_t, \pi) \propto \rho(s') \exp(-D(f(s'), \varphi(s_t, \pi)))$. Note that the definition of $\Psi(s_t, \varphi(s_t, \pi))$ may be expressed in terms of a KL:

$$\Psi(s_t, \varphi(s_t, \pi)) = \underset{\pi' \in \Pi}{\arg\min} \frac{1}{\overline{w}} \sum_{k=1}^{c} \gamma^{k-1} w_k D_{\text{KL}}(P_{\pi'}(s_{t+k}|s_t)||K(s_{t+k}|s_t, \pi)). \tag{42}$$

Therefore we have,

$$\frac{1}{\overline{w}} \sum_{k=1}^{c} \gamma^{k-1} w_k D_{\text{KL}}(P_{\Psi(s_t, \varphi(s_t, \pi))}(s_{t+k}|s_t)||K(s_{t+k}|s_t, \pi))$$

$$\leq \frac{1}{\overline{w}} \sum_{k=1}^{c} \gamma^{k-1} w_k D_{\text{KL}}(P_\pi(s_{t+k}|s_t)||K(s_{t+k}|s_t, \pi)). \tag{43}$$

By condition 8 we have,

$$\frac{1}{\overline{w}} \sum_{k=1}^{c} \gamma^{k-1} w_k D_{\text{KL}}(P_\pi(s_{t+k}|s_t)||K(s_{t+k}|s_t, \pi)) \leq \epsilon^2/8. \tag{44}$$

Jensen's inequality on the sqrt function then implies

$$\frac{1}{\overline{w}} \sum_{k=1}^{c} \gamma^{k-1} w_k \sqrt{2 D_{\text{KL}}(P_\pi(s_{t+k}|s_t)||K(s_{t+k}|s_t, \pi))} \leq \epsilon/2. \tag{45}$$

Pinsker's inequality now yields,

$$\frac{1}{\overline{w}}\sum_{k=1}^{c}\gamma^{k-1}w_k D_{\mathrm{TV}}(P_\pi(s_{t+k}|s_t)||K(s_{t+k}|s_t,\pi)) \le \epsilon/2. \tag{46}$$

Similarly Jensen's and Pinsker's inequality on the LHS of Equation 43 yields

$$\frac{1}{\overline{w}}\sum_{k=1}^{c}\gamma^{k-1}w_k D_{\mathrm{TV}}(P_{\Psi(s_t,\varphi(s_t,\pi))}(s_{t+k}|s_t)||K(s_{t+k}|s_t,\pi)) \le \epsilon/2. \tag{47}$$

The triangle inequality and Equations 46 and 47 then give us,

$$\frac{1}{\overline{w}}\sum_{k=1}^{c}\gamma^{k-1}w_k D_{\mathrm{TV}}(P_\pi(s_{t+k}|s_t)||P_{\Psi(s,\varphi(s,\pi))}(s_{t+k}|s_t))$$

$$\le \frac{1}{\overline{w}}\sum_{k=1}^{c}\gamma^{k-1}w_k \left( D_{\mathrm{TV}}(P_\pi(s_{t+k}|s_t)||K(s_{t+k}|s_t,\pi)) + D_{\mathrm{TV}}(P_{\Psi(s_t,\varphi(s_t,\pi))}(s_{t+k}|s_t)||K(s_{t+k}|s_t,\pi)) \right)$$

$$\le \epsilon, \tag{48}$$

as desired.

## C EXPERIMENTAL DETAILS

### C.1 ENVIRONMENTS

The environments for Ant Maze, Ant Push, and Ant Fall are as described in Nachum et al. (2018). During training, target $(x, y)$ locations are selected randomly from all possible points in the environment (in Ant Fall, the target includes a $z$ coordinate as well). Final results are evaluated on a single difficult target point, equal to that used in Nachum et al. (2018).

The Point Maze is equivalent to the Ant Maze, with size scaled down by a factor of 2 and the agent replaced with a point mass, which is controlled by actions of dimension two – one action determines a rotation on the pivot of the point mass and the other action determines a push or pull on the point mass in the direction of the pivot.

For the 'Images' versions of these environments, we zero-out the $x, y$ coordinates in the observation and append a low-resolution $5 \times 5 \times 3$ top-down view of the environment. The view is centered on the agent and each pixel covers the size of a large block (size equal to width of the corridor in Ant Maze). The 3 channels correspond to (1) immovable blocks (walls, gray in the videos), (2) movable blocks (shown in red in videos), and (3) chasms where the agent may fall.

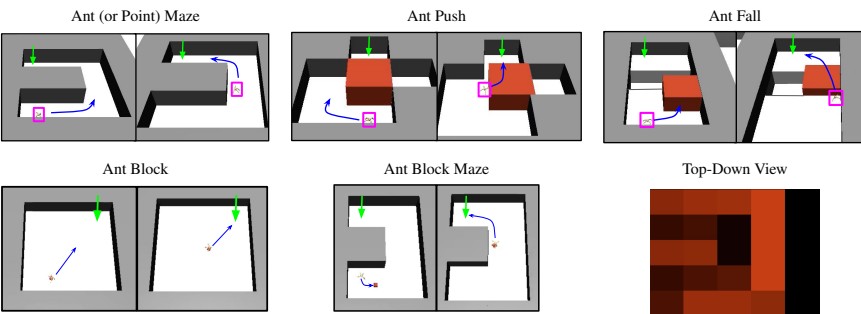

Figure 5: The tasks we consider in this paper. Each task is a form of navigation. The agent must navigate itself (or a small red block in 'Block' tasks) to the target location (green arrow). We also show an example top-down view image (from an episode on the Ant Maze task). The image is centered on the agent and shows walls and blocks (at times split across multiple pixels).

The Ant Block environment puts the ant in a $16 \times 16$ square room next to a $0.8 \times 0.8 \times 0.4$ small movable block. The agent is rewarded based on negative L2 distance of the block to a desired target location. During training, these target locations are sampled randomly from all possible locations. Evaluation is on a target location diagonally opposite the ant.

The Ant Block Maze environment consists of the same ant and small movable block in a $\supset$-shaped corridor. During training, these target locations are sampled randomly from all possible locations. Evaluation is on a target location at the end of the corridor.

## C.2 TRAINING DETAILS

We follow the basic training details used in Nachum et al. (2018). Some differences are listed below:

- We input the whole observation to the lower-level policy (Nachum et al. (2018) zero-out the $x, y$ coordinates for the lower-level policy).

- We use a Huber function for $D$, the distance function used to compute the low-level reward.

- We use a goal dimension of size 2. We train the higher-level policy to output actions in $[-10, 10]^2$. These actions correspond to desired deltas in state representation.

- We use a Gaussian with standard deviation 5 for high-level exploration.

- Additional differences in low-level training (e.g. reward weights and discounting) are implemented according to Section 5.

We parameterize $f_\theta$ with a feed-forward neural network with two hidden layers of dimension 100 using relu activations. The network structure for $\varphi_\theta$ is identical, except using hidden layer dimensions 400 and 300. We also parameterize $\varphi(s, \pi) := f_\theta(s) + \varphi_\theta(s, \pi)$. These networks are trained with the Adam optimizer using learning rate 0.0001.

## D OBJECTIVE FUNCTION EVALUATION

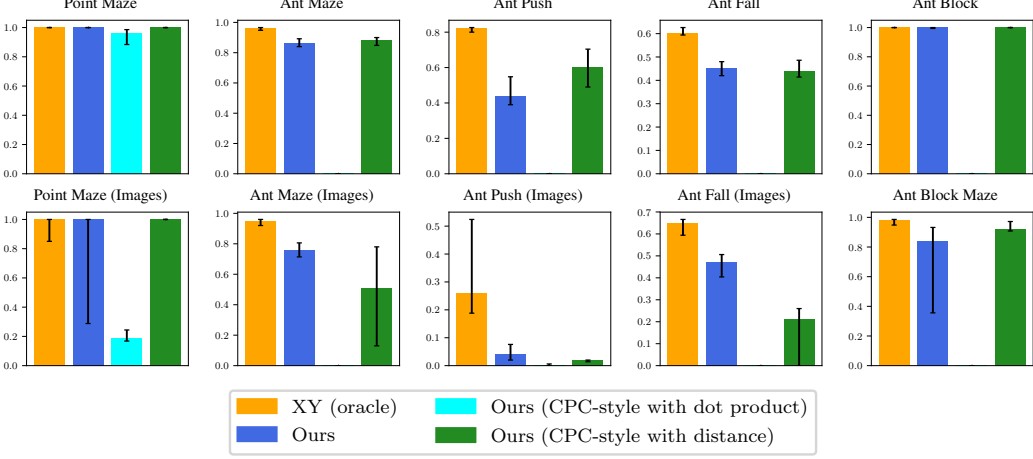

Figure 6: For our method, we utilize an objective based on Equation 8. The objective is similar to mutual information maximizing objectives (CPC; van den Oord et al. (2018)). We compare to variants of our method that are implemented more in the style of CPC. Although we find that using a dot product rather than distance function $D$ is detrimental, a number of distance-based variants of our approach may perform similarly.

# E $\beta$-VAE

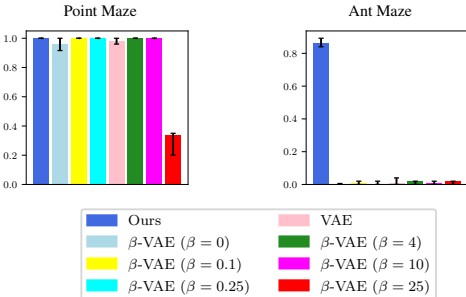

Figure 7: We provide additional results comparing to variants of $\beta$-VAE (Higgins et al., 2016). We find that even with this additional hyperparameter, the VAE approach to representation learning does not perform well outside of the simple point mass environment. The drawback of the VAE is that it is encouraged to reconstruct the entire observation, despite the fact that much of it is unimportant and possibly exhibits high variance (e.g. ant joint velocities). This means that outside of environments with high-information state observation features, a VAE approach to representation learning will suffer.

# F Generalization Capability

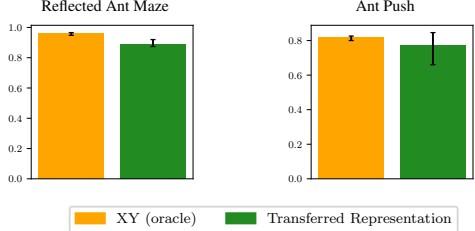

Figure 8: We evaluate the ability of our learned representations to transfer from one task to another. For these experiments, we took a representation function $f$ learned on Ant Maze, fixed it, and then used it to learn a hierarchical policy on a completely different task. We evaluated the ability of the representation to transfer to "Reflected Ant Maze" (same as Ant Maze but the maze shape is changed from '⊃' to '∩') and "Ant Push". We find that the representations are robust these changes to the environment and can generalize successfully. We are able to learn well-performing policies in these distinct environments even though the representation used was learned with respect to a different task.

# G Additional Qualitative Results

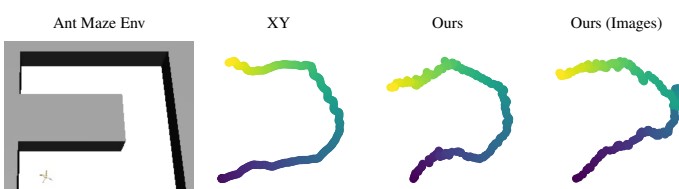

Figure 9: We replicate the results of Figure 2 but with representations learned according to data collected by a random higher-level policy. In this setting, when there is even less of a connection between the representation learning objective and the task objective, our method is able to recover near-ideal representations.

## H COMPARISON TO HIRO

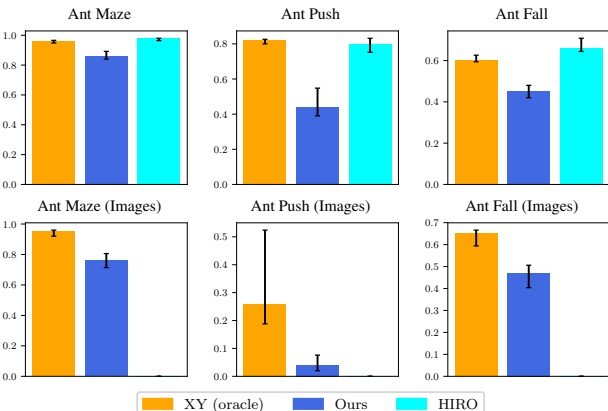

Figure 10: Results of our method compared to the original formulation of HIRO (Nachum et al., 2018). The representation used in the original formulation of HIRO is a type of oracle - sub-goals are defined as only the position-based (i.e., not velocity-based) components of the agent observation. In our own experiments, we found this method to perform similarly to the XY oracle in non-image tasks. However, when the state observation is more complex (images) performance is much worse.

---

**Algorithm 1** Representation learning for hierarchical RL.

---

**Input:** Replay buffer $\mathcal{D}$, number of training steps $N$, batch size $B$, parameterizations $f_\theta, \varphi_\theta, \pi_{\text{hi}}^\phi, \pi_{\text{lo}}^\phi$

**function** $EstLogPart(\widetilde{S}, s_t, a_{t:t+c-1})$
    *## Equation 15*
    Return $\log \frac{1}{|\widetilde{S}|} \sum_{\tilde{s} \in \widetilde{S}} \exp(-D(f_\theta(\tilde{s}), \varphi_\theta(s_t, a_{t:t+c-1})))$.
**end function**

**function** $CompLowReward(g, s_t, s', a_{t:t+c-1}, L)$
    *## Equation 18*
    Return $-D(f_\theta(s'), g) + D(f_\theta(s'), \varphi_\theta(s_t, a_{t:t+c-1})) + L$.
**end function**

**function** $CompReprLoss(s_t, s', a_{t:t+c-1}, \tilde{s}, L)$
    *## Equation 14*
    Compute attractive term $J_{\text{att}} = D(f_\theta(s'), \varphi_\theta(s_t, a_{t:t+c-1}))$.
    Compute repulsive term $J_{\text{rep}} = \exp(-D(f_\theta(\tilde{s}), \varphi_\theta(s_t, a_{t:t+c-1}))) - \text{stopgrad}(L)$.
    Return $J_{\text{att}} + J_{\text{rep}}$.
**end function**

**for** $T = 1$ **to** $N$ **do**

    Sample experience $g, s, a, r, s'$ and add to replay buffer $\mathcal{D}$ (Nachum et al., 2018).
    Sample batch of $c$-step transitions $\{(g^{(i)}, s_{t:t+c}^{(i)}, a_{t:t+c-1}^{(i)}, r_{t:t+c-1}^{(i)})\}_{i=1}^B \sim \mathcal{D}$.
    Sample indices into transition: $\{k^{(i)}\}_{i=1}^B \sim [1, c]^B$.
    Sample batch of states $\widetilde{S} = \{\tilde{s}^{(i)}\}_{i=1}^B \sim \mathcal{D}$.
    Estimate log-partitions $\{L^{(i)}\}_{i=1}^B = \{EstLogPart(\widetilde{S}, s_t^{(i)}, a_{t:t+c-1}^{(i)})\}_{i=1}^B$.

    *// Reinforcement learning*
    Compute low-level rewards $\{\tilde{r}^{(i)}\}_{i=1}^B = \{CompLowReward(g^{(i)}, s_t^{(i)}, s_{t+k^{(i)}}^{(i)}, a_{t:t+c-1}^{(i)}, L^{(i)})\}_{i=1}^B$.
    Update $\pi_{\text{lo}}^\phi$ (Nachum et al., 2018) with experience $\{(g^{(i)}, s_{t+k^{(i)}-1}^{(i)}, a_{t+k^{(i)}-1}^{(i)}, \tilde{r}^{(i)}, s_{t+k^{(i)}}^{(i)})\}_{i=1}^B$.
    Update $\pi_{\text{hi}}^\phi$ (Nachum et al., 2018) with experience $\{(g^{(i)}, s_{t:t+c}^{(i)}, a_{t:t+c-1}^{(i)}, r_{t:t+c-1}^{(i)})\}_{i=1}^B$.

    *// Representation learning*
    Compute loss $J = \frac{1}{B} \sum_{i=1}^B w_{k^{(i)}} \gamma^{k^{(i)}-1} CompReprLoss(k^{(i)}, s_t^{(i)}, s_{t+k^{(i)}}^{(i)}, a_{t:t+c-1}^{(i)}, \tilde{s}^{(i)}, L^{(i)})$.

    Update $\theta$ based on $\nabla_\theta J$.

**end for**

---

