# OpenReview forum: "Near-Optimal Representation Learning for Hierarchical Reinforcement Learning"
_ICLR.cc/2019/Conference_

### Official Review · AnonReviewer1 · 2018-11-02
**Interesting paper**

**Rating:** 7
**Confidence:** 5

**Review:**

The paper studies the problem of representation learning in the context of hierarchical reinforcement learning by building on the framework of  HIRO (Nachum et al. (2018)). The papers propose a way to handle sub-optimality in the context of learning representations which basically refers to the overall sub-optimality of the entire hierarchical polity with respect to the task reward. And hence, the only practical different from the HIRO paper is that the proposed method considers representation learning for the goals, while HIRO was directly using the state space.


I enjoyed reading the paper. The paper is very *well* written.

Experimental results:  The authors perform the series of experiments on various high dimensional mujoco env, and  show that the representations learned using the proposed method outperforms other methods (like VAEs, E2C etc), and can recover the controllable aspect of the agent i.e the x, y co-ordinate. This is pretty impressive result.

Some questions:

[1] Even though the results are very interesting, I'm curious as to how hard authors try to fit the VAE baseline. Did authors try using beta VAEs (or variants like InfoVAE) ?  Since the focus of the entire paper is about representation learning (as well as the focus of the conference), it is essential to make sure that baselines are strong. I would have suspected that it would have been possible to learn x,y co-ordinate in some cases while using improved version of VAE like beta VAE etc.

[2] One of the major intuitions behind sub-optimality is to learn representations that can generalize, as well as can be used for continual learning (or some variant of it!). This aspect is totally missing from the current paper. It would be interesting to show that the representations learned using the proposed method can transfer well to other scenarios or can generalize in the presence of new goals, or can be sample efficient in case of continual learning.

I think, including these results would make the paper very strong.  (and I would be happy to increase my score!).

---

> ### Author Response · Authors · 2018-11-16
> **Response**
>
> We thank the reviewer for the careful reading of the paper.  We are glad that the reviewer found the paper interesting and exceptionally well-written.  Our responses to the reviewer’s main feedback is below.  We hope that with the additional results, the reviewer will find the paper stronger.  Please let us know if this addresses your points, we would be happy to discuss further! But if your suggestions have been addressed, we would appreciate it if you would revise your review.
>
> For the VAE baseline, we performed tuning on the standard deviation of the Gaussian prior (finding std=10 to give best qualitative and quantitative results).  As the reviewer suggests, we have updated the paper (see Appendix E) to include results of VAE with varying beta (coefficient on KL prior).  Despite this additional hyperparameter tuning, results of the VAE are still lackluster.  The drawback of the VAE is that it is encouraged to reconstruct the entire observation, despite the fact that much of it is unimportant and possibly exhibits high variance (e.g. as in agent joint velocities).  This means that outside of environments with homogeneous and high-information state observation features, a VAE approach to representation learning will suffer.  And indeed, experimentally we find that the VAE can perform well on simple environments like point mass, which has only 6 observation dimensions, of which the velocities are often close to zero.
> Representations that generalize well are indeed very important.
>
> Empirically, we have followed the reviewer’s suggestions and have included in the paper (see Appendix F) additional results showing the generalization capability of the learned representations.  We take representations learned on one task (Ant Maze) and apply them to different tasks (a slight variant of Ant Maze and Ant Push).  Our representations are successful at training a policy on these distinct environments, thus showing the generalization capabilities of the learned representations across similar environments.
>
> Theoretically, we believe that our bounds may be extended to robust control settings, where representations should be robust to slight perturbations in transition dynamics (the KL divergence associated with the perturbations can be incorporated to our sub-optimality results).  A more in-depth theoretical treatment of generalization is a good open question for future work.

---

> > ### Comment · AnonReviewer1 · 2018-11-19
> > **Thanks!**
> >
> > Just for clarification for better generalization, If I understand it right, the transferred representation works better than directly learning on that env from scratch right ?
> >
> > Also, what are the results for HIRO ? where you are not doing representation learning for for the goals. I believe HIRO would more or less perform the same. It would be useful to add.

---

> > > ### Author Response · Authors · 2018-11-19
> > > **Additional clarifications**
> > >
> > > "If I understand it right, the transferred representation works better than directly learning on that env from scratch right ?"
> > >
> > > -- Yes you are correct, when comparing to Fig 3.  There are two main reasons for this: (1) The results in Fig 3 use representations learned *online* and from scratch concurrently while learning the HRL policy.  This can cause difficulties in training compared to using a fixed, pre-trained representation.  (2) One important component necessary to solve the "Ant Push" task is exploration along the x-axis, and this can be exacerbated when learning representations concurrently (if there is not much experience of exploration along the x-axis, the learning of representations in these regions will suffer).  This is somewhat remedied by transferring pre-trained representations from the "Ant Maze" task, where the exploration problem is less pronounced and thus representations are learned well along most of the x-axis.
> > >
> > > "Also, what are the results for HIRO ?"
> > >
> > > -- We have now included results for HIRO (at least those which we have immediately available) in Appendix H.  We note that the representation used in Nachum et al 2018 is a type of oracle - in fact, they define sub-goals as only the position-based (i.e. not velocity-based) components of the agent observation (compare this to the poor results of our “whole obs” baseline, which uses both position and velocity observations).  In our own experiments, we found the HIRO method used in Nachum et al 2018 to perform similarly to the X-Y oracle in non-image tasks and similarly to the "whole obs" baseline in image tasks, and thus opted to show only a single “oracle” baseline and a single "whole obs" baseline in the main text.

---

> > > > ### Comment · AnonReviewer1 · 2018-11-20
> > > > **Thanks!**
> > > >
> > > > I have increased my score.
> > > >
> > > > I like the presentation of the paper, I like empirical results.
> > > >
> > > > ==================================
> > > >
> > > > This comment has nothing much to do with the paper.
> > > >
> > > > One thought though. I wonder, if the "sub-optimality" could (also) give rise to "state-abstractions" (state abstractions as in RL, where you reduce your MDP problem by converting your dense MDP to somewhat sparse MDP). The intuition is, that the states which have more or less same degree of sub-optimality can be mapped to same "abstract state".
> > > >
> > > > I am curious as to what authors have to say in this regard.

---

> > > > > ### Author Response · Authors · 2018-11-21
> > > > > **State Abstractions**
> > > > >
> > > > > Similar notions do exist in the state abstraction literature.  The reviewer may already be aware of Abel, et al. "Near Optimal Behavior via Approximate State Abstraction."  In this work, the authors explore an MDP abstraction based on a many-to-one state abstraction mapping.  A notion of sub-optimality of the state abstraction is defined, and it is similar in spirit to the notion presented in our work.  One notable theoretical finding in Abel, et al. (slightly different from the reviewer's suggestion) is that states with similar Q*-values should be mapped to the same abstract state.  Grouping states with similar sub-optimalities (in our sense of the word) may be possible, but harder to approach theoretically due to the dependence on \Psi, which could potentially be very poor.
> > > > >
> > > > > In the Abel, et al. paper there are several other theoretical insights. Unfortunately, the application of such theory to difficult control tasks is so far lacking. Many of the notions of a good state abstraction are difficult to convert to practical representation learning objectives.  Moreover, the assumption of a finite, discrete, abstract state space may be very restrictive and further hamper learning.  Future work in this area could be very impactful.

---

### Official Review · AnonReviewer3 · 2018-11-02
**The first clear formulation of goal-oriented HRL**

**Rating:** 9
**Confidence:** 5

**Review:**

The problem setting considered in this paper is that of the recent wave of "goal-conditioned" formulations for hierarchical control in Reinforcement Learning. In this problem, a low-level controller is incentivized to reach a goal state designated by a higher-level controller. This goal is represented in an abstract (embedding) multi-dimensional vector space. Establishing "closeness to goal" entails the existence of some distance metric (assumed to be given an fixed) and a function $f$ which can project states to their corresponding goal representation. The "representation learning" problem referred to by the authors pertains to this function. The paper is built around the question: how does the choice of $f$ affects the expressivity of the class of policies induced in the lower level controller, which in turn affects the optimality of the overall system. The authors answer this question by first providing a bound on the loss of optimality due to the potential mismatch between the distribution over next states under the choice of primitive actions produced by a locally optimal low-level controller. The structure of the argument mimics that of model compression methods based on bisimulation metrics. The model compression here is with respect to the actions (or behaviors) rather than states (as in aggregation/bismulation methods). In that sense, this paper is a valuable contribution to the more general problem of understanding the nature of the interaction between state abstraction and temporal abstraction and where the two may blend (as discussed by Dietterich and MAXQ or Konidaris for example). Using the proposed bounds as an objective, the authors then derive a gradient-based algorithm for learning a better $f$. While restricted to a specific kind of temporal abstraction model, this paper offers the first (to my knowledge) clear formulation  of "goal-conditioned" (which I believe is an expression proposed by the authors) HRL fleshed out of architectural and algorithmic considerations. The template of analysis is also novel and may even be useful in the more general SMDP/options perspective. I recommend this paper for acceptance mostly based on this: I believe that these two aspects will be lasting contributions (much more than the specifics of the proposed algorithms).


# Comments and Questions

It's certainly good to pitch the paper as a "representation learning" paper at a Representation Learning conference, but I would be careful in using this expression too broadly. The term "representation" can mean different things depending on what part of the system is considered. Representation learning of the policies, value functions etc. I don't have specific recommendations for how to phrase things differently, but please make sure to define upfront which represention you are referring to.

Representation learning in the sense of let's say Baxter (1995) or Minsky (1961) is more about "ease of learning" (computation, number of samples etc) than "accuracy". In the same way, one could argue that options are more about learning more easily (faster) than for getting more reward (primitive options achieve the optimal). Rather than quantifying the loss of optimality, it would be interesting to also understand how much one gains in terms of convergence speed for a given $f$ versus another. I would like to see (it's up to you) this question being discussed in your paper. In other words, I think that you need to provide some more motivation as to why think the representation learning of $f$ should be equated with the problem of maximizing the return. One reason why I think that is stems from the model formulation in the first place: the low-level controller is a local one and maximizes its own pseudo-reward (vs one that knows about other goals and what the higher level controller may do). It's both a feature, and limitation of this model formulation; the "full information" counterpart also has its drawbacks.

A limitation of this work is also that the analysis for the temporally extended version of the low-level controller is restricted to open-loop policies. The extension to closed-loop policies is important. There is also some arbitrariness in the choice of distance function which would be important to study.

Relevant work (it's up to you to include or not):

- Philip Thomas and Andrew Barto in "Motor Primitive Discovery" (2012) also talk about options-like abstraction in terms of compression of action. You may want to have a look.

- Still and Precup (2011) in "An information-theoretic approach to curiosity-driven reinforcement learning" also talk about viewing actions as "summary of the state" (in their own words). In particular, they look at minimizing the mutual information between state-action pairs.

- More generally, I think that the idea of finding "lossless" subgoal representations is also related to ideas of "empowerment" (the line of work of Polani).

---

> ### Author Response · Authors · 2018-11-16
> **Response**
>
> We thank the reviewer for the careful reading of the paper.  We are glad that the reviewer believes the paper to be a significant contribution to the field of hierarchical RL.  Our responses to the reviewer’s comments are below.  We are happy to discuss further if the reviewer has additional comments.
>
> “make sure to define upfront which representation you are referring to”
> -- We have updated the introduction to make this clear.
>
> “I think that you need to provide some more motivation as to why think the representation learning of $f$ should be equated with the problem of maximizing the return.“
> -- The issue we believed to be most pressing when using representations in HRL is that of expressibility.  Although the optimal policy may no longer be expressible, one can still hope to approximately express the optimal policy.  To make this notion more approachable, we defined sub-optimality in terms of state values, and this leads to the analysis presented in the paper.  It is also more practical to define sub-optimality in this way for the simple reason that a simple mathematical formulation of this sub-optimality is easy to express, compared to notions like the reviewer’s suggested “ease of learning,” which is harder to quantify in the context of deep RL.  We have updated some of the phrasing in Sec 3 to make this clearer.
>
> “A limitation of this work is also that the analysis for the temporally extended version of the low-level controller is restricted to open-loop policies.”
> -- We would like to clarify that the theoretical statements (Theorem 3 and Claim 4) apply to any set of candidate policies \Pi, including possibly closed-loop policies.  We have updated the paper to make this clear.
>
> “Relevant work (it's up to you to include or not):“
> -- Thank you for these references.  We will add them to the paper.

---

### Official Review · AnonReviewer4 · 2018-11-12
**Relevant topic, interesting formulation, not clear what is the benefit of "good" representation**

**Rating:** 8
**Confidence:** 3

**Review:**

-- Summary --

The authors proposes a novel approach in learning a representation for HRL. They define the notion of sub-optimality of a representation (a mapping from observation space to goal space) for a goal-conditioned HRL that measure the loss in the value as a result of using a representation. Authors then state an intriguing connection between representation learning and bounding the sub-optimality which results in a gradient based algorithm.

-- Clarity --

The paper is very well written and easy to follow.

-- novelty --
To the best of my knowledge, this is the first paper that formalizes a framework for sub-optimality of goal-conditioned HRL and I think this is the main contribution of the paper, that might have lasting effect in the field. This works mainly builds on top of Nachum et al 2018 for Data Efficient HRL.

-- Questions and Concerns --

[1] Authors discussed the quality of the representation by comparing to some near optimal representation (x,y) in section 7. My main concern is how really “good” is the representation?, being able to recover a representation like x,y location is very impressive, but  I think of a “good” representation as either a mapping that can be generalized or facilitate the learning (reducing the sample complexity). For example Figure 3 shows the performance after 10M steps, I would like to see a learning curve and comparison to previous works like Nachum et al 2018 and see if this approach resulted in a better (in term of sample complexity) algorithm than HIRO. Or an experiment that show the learned representation can be generalized to other tasks.

[2] What are author's insight toward low level objective function? (equation 5 for example) I think there could be more discussion about equation 5 and why this is a good objective to optimize. For example third term is an entropy of the next state given actions, which will be zero in the case of deterministic environment, so the objective is incentivizing for actions that reduce the distance, and also has more deterministic outcome (it’s kind of like empowerment - klyubin 2015). I’m not sure about the prior term, would be great to hear authors thoughts on that. (a connection between MI is partially answering that question but I think would be helpful to add more discussion about this)

[3] Distance function : In section C1 authors mention that they used L2 distance function for agent’s reward. That’s a natural choice, although would be great to study the effect of distance functions. But my main concern is that author's main claim of a good representation quality is the fact that they recovered similar to XY representation, but that might simply be an artifact of the distance function, and that (x,y) can imitate reward very well. I am curious to see what the learned representation would be with a different distance function.

--
I think the paper is strong, and interesting however i’d like to hear authors thoughts on [2], and addressing [1] and [3] would make the paper much stronger.

---

> ### Author Response · Authors · 2018-11-16
> **Response**
>
> We thank the reviewer for the valuable feedback. We are glad that the reviewer found the paper well-written and exceptionally novel.  Our responses to the reviewer’s main points are below, and we hope with these clarifications and additional results, the reviewer will find the paper much stronger.  Please let us know if this addresses your points, we would be happy to discuss further! But if your suggestions have been addressed, we would appreciate it if you would revise your review.
>
> [1]
> “My main concern is how really “good” is the representation?”
>
> The results in Figure 3 are our most significant.  They show that the sub-optimality theory we develop holds up in practice: We are able to learn representations (in an online fashion, with no additional supervision) which allow for learning a well-performing hierarchical policy, and this ability is significantly better than competing methods.
>
> “I would like to see… an experiment that shows the learned representation can be generalized to other tasks.”
>
> We have followed the reviewer’s suggestions and have included in the paper (see Appendix F) additional results showing the generalization capability of the learned representations.  We take representations learned on one task (Ant Maze) and apply them to different tasks (a slight variant of Ant Maze and Ant Push).  Our representations are successful at training a policy on these distinct environments, thus showing the generalization capabilities of the learned representations across similar environments.
>
> “I think of a “good” representation as a mapping that can ... facilitate the learning (reducing the sample complexity). I would like to see a learning curve and comparison to previous works like Nachum et al 2018”
>
> The representation used in Nachum et al 2018 is a type of oracle - in fact, they define sub-goals as only the position-based (i.e. not velocity-based) components of the agent observation (compare this to the poor results of our “whole obs” baseline, which uses both position and velocity observations).  In our own experiments, we found the HIRO method used in Nachum et al 2018 to perform similarly to the X-Y oracle in non-image tasks and similarly to the "whole obs" baseline in image tasks, and thus opted to show only a single “oracle” baseline and a single "whole obs" baseline.
>
> As for facilitating learning: When one uses HRL, much of the ease-of-learning is induced automatically - e.g., the higher-level controller operates and receives rewards at a much lower temporal frequency, hence learning and exploration improves.  On the other hand, the use of a lower-level goal-conditioned policy as an interface between the higher-level controller and the environment introduces a potential issue in how expressible the high-level controller can be.  This is the issue we focused on in our paper.  We view HIRO (Nachum et al 2018) as a work showing that with HRL, one can learn much better and faster than shallow policies; our work then shows that in general settings, where a goal representation may not be easy to handcraft (e.g. images), one can still learn provably near-optimal representations.  And indeed, it is important to note that our presented results are better than the shallow baselines (and several of the hierarchical baselines) evaluated in the HIRO paper, showing that we can get good HRL performance on these difficult tasks while incorporating absolutely zero prior knowledge into the learning algorithm.
>
> [2]
> “What are author's insight toward low level objective function? (equation 5 for example)”
>
> The form of Equation 5 (or Equation 9) is particularly chosen so that it corresponds to a KL (this is crucial for the sub-optimality proof in the Appendix).  Specifically, Equation 5 defines the low-level objective as a negative KL between P(s’| s, a) and a distribution U(s’| s, g) proportional to \rho(s’) * exp(-D(f(s’), g)).  We have updated the paper to make this observation clear.
>
> “... third term is an entropy of the next state given actions, which will be zero in the case of deterministic environment, so the objective is incentivizing for actions that reduce the distance, and also has more deterministic outcome.”
>
> As the KL insight above makes clear, the objective does not necessarily encourage deterministic outcomes.  Instead, it encourages the lower-level policy to find actions whose next-state distribution is similar to U(s’| s, g) (which is stochastic in general).

---

> > ### Author Response · Authors · 2018-11-16
> > **Response (cont)**
> >
> > [3]
> > “My main concern is that author's main claim of a good representation quality is the fact that they recovered similar to XY representation, but that might simply be an artifact of the distance function, and that (x,y) can imitate reward very well.”
> >
> >
> > We would like to emphasize that the representation learning objective is agnostic to task reward; i.e., unlike possibly other RL representation learning objectives, there is no reward prediction based on learned representations.
> >
> > The only way in which task reward may affect learning is by the distribution of states/goals induced by a higher-level controller trained to maximize reward.  Thus, to address the reviewer’s concerns, we have repeated the experiment in Figure 2 for our proposed representation learning objective (see Appendix G), but learned with a uniformly random higher-level controller.  We find that the learned representations can still recover representations similar to x-y coordinates.

---

> > > ### Comment · AnonReviewer4 · 2018-11-22
> > > **Thanks!**
> > >
> > > Helpful response!

---

### Author Response · Authors · 2018-10-02
**Typographical Errors**

We would like to make readers aware of a few typos in Appendix B.  Namely:

-- Eq 42 overloads the use of \pi.  The equation should be changed to be an argmin over \pi'\in\Pi.  Accordingly, the use of P_\pi in the KL should be changed to P_{\pi'}.  The other argument K(-|-,\pi) of the KL remains as-is.

-- The LHS of Eq 42 should have P_{\Psi(s_t, \varphi(s_t, \pi))} in the KL (the current form does not include the 't' subscripts and is missing a closing parens).

---

### Public Comment · ~Olivier_Sigaud1 · 2018-12-06
**More about learning, please**

This is an important paper which makes several strong points and may get an oral presentation, so we can ask the authors to make it even better.

Actually, some aspects need to be elaborated further about the learning part.

If I understood correctly, the algorithm is starting with an empty replay buffer and a random "goal representation" network.
Then the (initially random) hierarchical policy using this random goal representation is used to fill the replay buffer, from which samples are drawn to improve simultaneously the representation and the hierarchical policies. Is this correct?

The main point is then the following:
- there must be some condition (a form of iid-ness?) on the content of the replay buffer so that the goal representation gets properly improved (and the hierarchical policy too),
- there must be some condition on the hierarchical policy (some exploration property?) so that the replay buffer gets properly filled
- there must be some condition on the goal representation so that the hierarchical policy does something valuable.

Off-policy deep RL algorithms using a replay buffer like DDPG and TD3 are already known to be rather unstable and sensitive to hyper-parameter setting. The three conditions above make me feel that your algorithm might be even worse in that respect. You obtained good results, but can you elaborate on how hard it was to get them? Do you believe one can get some guarantees about using this algorithm in practice?

By the way, to get a better grasp on these questions, very few is specified about learning in your paper:
- are you using TD3, as you were doing in the HIRO paper?
- can you say a word about hyper-parameter search, final hyper-parameter setting? None is specified (replay buffer size, Huber function factor, learning rates...)
- can you specify the computational effort (wall time, number of CPUs/GPUs...) it took to get your results?

Side points:
- It might be a good idea to give a name to your algorithm, to facilitate further references by the community
- typo: last line of Algo 1, there is a "=0" which should probably be removed
- HIRO is cited under Nachum et al. 2018a and 2018b, the paper appears twice in the references

---

> ### Author Response · Authors · 2018-12-10
> **Response**
>
> Thanks for the close reading of the paper and the helpful feedback.  Answers to your concerns are below.  Let us know if you have additional questions!
>
> "If I understood correctly, ... Is this correct?"
> Yes, your understanding is correct.  To us, this was a relatively straightforward way to translate the theoretical results to a practical learning algorithm.  Nevertheless, as you correctly note, this practical approximation to the theory does not carry with it any strong theoretical guarantees.  The main difference between the theory and the practice is that the theory requires the objective to be optimized w.r.t. a supremum.  In practice, this supremum is loosely translated to an expectation over the replay buffer.  Despite the loose translation, we found the empirical results of our practical implementation to be strong.  All experiments used the same hyperparameters, most of which carried over from the original HIRO formulation.
>
> "Do you believe one can get some guarantees about using this algorithm in practice?"
> This question is one which is good for future work to explore.  One can probably replace the supremum in the theory with an expectation over a specific policy, while at the same time weakening the sub-optimality result from being w.r.t. the optimal policy to being w.r.t. a slightly higher-reward policy (in the style of TRPO).
>
> Hyperparameters:
> We provide experimental details in Appendix C.2.  Much of the original HIRO setup is maintained.  The code we use is also available as open-source (link not included to maintain anonymity, although you may easily search for it online).
>
> Computation:
> Each training run was run on a single machine with ~16 CPUs and took about a day to complete.

---

### Meta-Review · Area_Chair1 · 2018-12-14
**Strong paper on hierarchical RL with very strong reviews from people expert in this subarea that I know well.**

**Confidence:** 4
**Recommendation:** Accept (Poster)

**Metareview:**

Strong paper on hierarchical RL with very strong reviews from people expert in this subarea that I know well.